# Zoonotic Microparasites in Invasive Black Rats (*Rattus rattus*) from Small Islands in Central Italy

**DOI:** 10.3390/ani13203279

**Published:** 2023-10-20

**Authors:** Stefania Zanet, Flavia Occhibove, Dario Capizzi, Sara Fratini, Francesca Giannini, Avner Dan Hoida, Paolo Sposimo, Flaminia Valentini, Ezio Ferroglio

**Affiliations:** 1Department of Veterinary Sciences, University of Turin, Largo Paolo Braccini, 2, 10095 Grugliasco, Italy; stefania.zanet@unito.it (S.Z.); avnerhoida@gmail.com (A.D.H.); flaminia.valentini@unito.it (F.V.); ezio.ferroglio@unito.it (E.F.); 2Directorate for Natural Capital, Latium Region, Parks and Protected Areas, Viale del Tintoretto 432, 00142 Rome, Italy; dcapizzi@regione.lazio.it; 3Dipartimento di Biologia, Università di Firenze, Via Madonna del Piano 6, 50019 Sesto Fiorentino, Italy; sara.fratini@unifi.it; 4Parco Nazionale Arcipelago Toscano, Loc. Enfola, 57037 Portoferraio, Italy; giannini@islepark.it; 5Nature and Environment Management Operators SRL (NEMO), Piazza Massimo D’Azeglio 11, 50121 Florence, Italy; sposimo@nemoambiente.com

**Keywords:** *Anaplasma* spp., *Babesia* spp., black rat, *Leishmania* spp., *Neospora caninum*, *Toxoplasma gondii*, zoonosis

## Abstract

**Simple Summary:**

Invasive species negatively affect native populations through predation, competition, and the potential introduction of health threats, such as parasites. Black rats (*Rattus rattus*) are among the worst invaders of islands, and a significant source of parasites infecting humans and other animals. This study conducted a screening for zoonotic and veterinary-relevant microparasites in wild rats from small islands in central Italy, including the Pontine Islands and Pianosa, where the primary hosts of the selected parasites were either absent or scarce. The aim was to investigate the potential role of rats as their host. Rats were kill-trapped and molecular analyses were performed on different tissues to identify microparasite presence. Results confirm that invasive species such as rats may contribute to an elevated parasitological threat to local wildlife and human communities in specific ecosystems. Notably, we documented the first record of *Babesia divergens*, typically associated with cattle and wild ungulates, in wild rats. Additionally, we confirmed the presence of *Leishmania infantum* on an island without dogs, which have traditionally been considered the primary hosts. Our study helps to document parasite distribution and interactions between parasites and introduced invasive hosts, and represents useful knowledge to inform public health and wildlife management policy.

**Abstract:**

Invasive species have a detrimental impact on native populations, particularly in island ecosystems, and they pose a potential zoonotic and wildlife threat. Black rats (*Rattus rattus*) are invasive species that disrupt native flora and fauna on islands and serve as potential competent reservoirs for various pathogens and parasites. Microparasites screening was conducted in rat populations from small islands in central Italy (the Pontine Islands and Pianosa) with the aim of assessing the role of rats in maintaining infections, particularly in cases where key reservoir hosts were scarce or absent. We focused on microparasites of zoonotic and veterinary relevance. A total of 53 rats was kill-trapped and target tissues were analysed with molecular techniques. We observed the absence or very low prevalence of *Anaplasma* spp., while *Babesia* was found in rats from all locations, marking the first recorded instance of *Babesia divergens* in wild rats. Data from Pianosa strongly suggest the presence of an autochthonous *Leishmania infantum* cycle in the Tuscan archipelago islands. *Neospora caninum* was absent from all islands, even in areas where dogs, the main reservoirs, were present. *Toxoplasma gondii* was only recorded on the Pontine Islands, where genotyping is needed to shed light on infection dynamics. This study confirms that invasive species, such as rats, may be responsible for maintaining an increased parasitological threat to fauna and human communities in certain ecosystems.

## 1. Introduction

Islands, and especially small islands, are privileged sites for epidemiological investigations. This is because of their isolation, the uniqueness of their host–parasite associations, and the generally detailed information regarding their biogeography. Furthermore, island ecosystems are especially prone to the negative consequences of alien species introductions [1,2]. Alien and invasive species may include parasites able to infect native species, leading to the establishment of new epidemiological dynamics [3]. In this context, invasive animal populations may also represent a threat to human health, either introducing new zoonotic parasites [4] or amplifying existing zoonotic threats [5,6]. Thus, increased knowledge related to parasitism and processes driving parasite distribution among native and introduced fauna is a conservation and public health issue. 

Black rats (*Rattus rattus* Linnaeus, 1758) are known to be among the worst invaders of island ecosystems, negatively affecting flora, fauna and ecosystem functions [7,8]. This species displays a high reproductive potential, opportunistically exploits a wide range of food sources, and lacks significant predators as well as competitors [8]. For example, they heavily predate upon seabirds at all life stages including the eggs, nestlings and adults [9], and of a large range of other vertebrate, invertebrate and plant taxa [2,10,11]. This behaviour has been observed on Mediterranean islands [12], including Italian ones, where the black rat represents by far the most widespread terrestrial mammal, occurring on about 80% of the islands [13,14]. In addition, rats are an important source of pathogens for humans [15,16], targeted by several eradication programs, including on the small islands of central Italy, the region of focus in this study [17,18]. Rodent-borne pathogens comprise some of the most important emergent and re-emergent zoonoses. Rodents are one of the taxa with the highest zoonotic potential worldwide, representing a significant public health concern [19,20,21], most likely due to their life traits (e.g., early sexual maturity, high reproductive rate, large litters) [22]. In Italy, rats have been confirmed to be competent hosts of a wide range of potentially zoonotic pathogens and parasites: Hantavirus (cases of haemorrhagic fever with renal syndrome—HFRS); *Cryptosporidium parvum* (cryptosporidiosis); *Leishmania infantum* (leishmaniosis); *Leptospira interrogans* (leptospirosis); *Rickettsia conorii* (Mediterranean spotted fever); *Rickettsia typhi* (murine typhus); *Salmonella enteridis* (salmonellosis); *Streptobacillus moniliformis* and *Spirillum minus* (rat-bite fever); *Toxoplasma gondii* (toxoplasmosis) [23]. 

Hence, invasive rats may be responsible for maintaining an increased parasitological threat to fauna and human communities on islands, as observed by the high prevalence and diversity of helminths harboured by rats on Christmas Island (Australia) [24]. On Montecristo island (Italy), rats were found to be a competent reservoir of *L. infantum*, able to maintain the epidemiological cycle in the absence of the main reservoir host (i.e., dog) [25]. *T. gondii* sampled from invasive black rats on a Brazilian island showed high genetic variability, suggesting that adaptations to new environmental conditions and hosts may also lead to variations in virulence and pathogenicity [26]. Understanding rat-associated pathogens and parasites is crucial for disease-control policy, especially on islands, where local epidemiological dynamics are altered by invasive rat populations and their parasites. Parasitological surveys of invasive species are also recommended in eradication frameworks; invasive parasites may survive the eradication of their invasive hosts, representing a potential threat to native populations [27].

*Babesia* spp. and *Theileria* spp. are protozoan parasites transmitted mainly by tick vectors, with a tropism for erythrocytes and/or leukocytes of a wide range of domestic and wildlife species [28]. *Babesia* spp. are among the most common parasites found in mammals’ blood [29], with reports of zoonotic infections not only from the most common *B. divergens*, but also from the rodent specific *B. microti* [30]. Some species of *Theileria* spp. are highly pathogenic to cattle and may cause significant mortality, while other are considered to be less pathogenic, but may cause clinical signs in stressed, immunodeficient, or malnourished individuals [28,31,32]. Although piroplasmosis is a frequent and disrupting disease in domestic animals [33], many uncertainties remain regarding its epidemiological dynamics in Ixodid tick vectors and vertebrate hosts, especially concerning wildlife host–vector–parasite dynamics [34,35,36]. 

Members of the genera *Ehrlichia* and *Anaplasma* are obligate intracellular bacteria, targeting host granulocytes or monocytes. Transmitted by Ixodid ticks, they are responsible for rickettsiosis, an infection of veterinary importance in livestock and companion animals [37]. Following the reorganisation of the Anaplasmatacea family, *E. equi* and *E. phagocytophila*, previously known as agents of human granulocytic ehrlichiosis, are now collectively referred to as *A. phagocytophilum* [38]. However, not all strains are pathogenic for humans, and the epidemiological cycle is still poorly understood due to a large number of potential reservoir hosts, the broad distribution and extensive niches of tick vector species and the various bacterial genotypes identified [39]. 

*L. infantum* (parasitic protozoan of the order Trypanosomatida) is the cause of leishmaniosis, a severe disease of domestic and wild animals transmitted by *Phlebotomus* sandflies [40]. Cells of the phagocytic mononuclear system represent the preferential niche of this parasite. Although widespread and highly prevalent in many areas, the transmission role of different mammalian host species of *Leishmania* spp. is still unclear [41]. It has been suggested that the black rat is a competent reservoir host for *L. infantum*, since recent studies support the hypothesis of sylvatic cycles of leishmaniosis independent from dogs; yet, the latter are still considered to be the main reservoir of infection [25,42]. 

*T. gondii* and *Neospora caninum* are coccidian parasites able to infect a wide range of warm-blooded vertebrates. The definitive hosts of the first are felids, while canids act as definitive hosts for the latter [43]. Small mammals, and rodents in particular, are essential intermediate hosts of both protozoa, representing a source of infections for final and paratenic hosts [44,45]. These parasites have a zoonotic, veterinary, and economic relevance. *T. gondii* is one of the most widespread zoonotic parasites, and it may cause clinical signs in humans, as well as domestic and wildlife species [26,46]. *N. caninum* is one of the primary causes of abortion in bovines. In addition, it causes reproductive disruption in small ruminant species and clinical manifestations in dogs [47]. 

In the context of eradication campaigns on small islands in central Italy (the Pontine Islands and Pianosa—Tuscan archipelago), an epidemiological screening was conducted to ascertain the presence and the prevalence of microparasites of zoonotic significance. The aim was to evaluate the potential reservoir role of rats in sustaining infections of directly borne and vector-borne microparasites in areas where the key reservoir species were absent or limited. Islands included in this study present different degrees of isolation, and different wildlife and domestic species communities. The Pontine Islands are inhabited and considered a popular tourist destination. Pianosa has not been permanently inhabited for the past 20 years, with no farming activities and a high degree of isolation. Zoonotic diseases are currently a significant threat to human health due to anthropogenic environmental changes. For example, in the context of vector-borne diseases, these changes alter vector distribution and the frequency of infection [48], and rodents play a major role in their life cycle [15,49]. Thus, microparasites of zoonotic and veterinary relevance were selected for screening purposes: *Babesia* spp., *Theileria* spp., *Anaplasma* spp., and *Ehrlichia* spp. (all the previously mentioned parasites transmitted by ticks); *L. infantum* (transmitted by sand-flies); and the directly transmitted *T. gondii* and *Neospora caninum* coccidia.

## 2. Materials and Methods

### 2.1. Study Area

The study area included three small islands in the Pontine Islands (Ponza, Ventotene, and Palmarola), and the small island of Pianosa (Tuscan archipelago), all located in the Mediterranean sea off the coast of central Italy (Figure 1). 

Ponza is the largest of the Pontine Islands, measuring 7.5 km^2^, and it is located 33 km south of Cape Circeo. It has been inhabited since the Neolithic, housing a current population of around 3300 people. It is characterised by maquis shrubland type vegetation cover, hosting several Mediterranean faunal species, notably resident and migratory birds. Ventotene has an area of 1.75 km^2^ and is located 33 km off the coast of the town of Gaeta; the current permanent human population is less than 1000 people. This island was originally covered by woodlands dominated by *Quercus ilex* together with maquis shrubland. Although the island is still a significant hotspot for migratory birds—e.g., Scopoli’s shearwater (*Calonectris diomedea*) and the Mediterranean shearwater (*Puffinus yelkouan*) —, the original habitats have been subjected to major anthropogenic disturbance. Palmarola is the third largest island in the archipelago, after Ponza and Ventotene, respectively, with an area of 1.36 km^2^. It is located 10 km west of Ponza and it entirely designated as a site of community importance (SIC) under the Habitat Directive (92/43/CEE), especially because of the significance of its bird species diversity (e.g., yellow-legged gull (*Larus michahellis*), peregrine falcon (*Falco peregrinus*), great cormorant (*Phalacrocorax carbo*)). While not permanently inhabited by humans, it hosts an established population of alien goats introduced in the 1990s, which have damaged the pristine habitats. Similarly to the goats, another invasive alien species has been disturbing native ones, not only on Palmarola but also on Ponza and Ventotene: the black rat (*Rattus rattus*). Species of wild terrestrial mammals are all extinct from these islands, while domestic species including dogs, cats, and rabbits are present. 

Due to their naturalistic value, these islands were elected for black rat eradication. The European LIFE project “PonDerat” (LIFE14 NAT/IT/000544) was implemented, among other biosecurity and conservation aims, to eradicate rats and other invasive species from the Pontine Islands. Globally, the black rat has been recognised to negatively affect native species and their habitats [12,50], and locally it greatly impacts seabird populations, whose chicks are heavily predated upon [17].

Pianosa is an unpopulated island, part of the Tuscan archipelago, located 14 km southwest of the island of Elba. Only a few people per day are allowed to set foot on it, as it is a protected natural area (Tuscan Archipelago National Park). Rarely, small groups of tourists can stay overnight at the small inn managed by volunteers and convicted prisoners who reside on Elba. Pianosa is an island of 10.25 km^2^ which was used as a penal colony from 1856 until 1998. The vegetation is characterised by rockrose (*Cistus monspeliensis*), rosemary (*Rosmarinus officinalis*), germander (*Teucrium fruticans* and *T. flavum*), and bushy varieties of mastic (*Pistacia lentiscus*) and olive (*Olea europea*). Wildlife species includes several species of birds (resident and migratory) and bats of significant conservation value as well as the brown hare (*Lepus europaeus*); invasive alien species are represented by the black rat (*Rattus rattus*) and the house mouse (*Mus musculus*). While the penal colony was in operation, livestock was present.

Similarly to the Pontine, Pianosa and other islands in the same archipelago (designated as Natura 2000 sites) were prioritised for rat eradication (LIFE13 NAT/IT/000471—RESTO CON LIFE “Island conservation in Tuscany, restoring habitat not only for birds”). Indeed, black rats have been recognised to be extremely harmful to the ecosystems of this island [8,14].

### 2.2. Rat Sampling, DNA Extraction and PCR Analysis

Black rats were trapped using TRex snap rat traps (Bell Laboratories Inc., Madison, WI, USA) as part of preliminary work for their eradication in the context of the abovementioned LIFE projects (ethical approval and standardised methodologies are detailed in project reports (PonDerat (LIFE14 NAT/IT/000544) http://www.ponderat.eu/documenti/pagine/life14_nat_it_000544_definitivo_2.PDF [accessed on 25 September 2022]; RESTO CON LIFE (LIFE13 NAT/IT/000471) https://www.restoconlife.eu/wordpress/wp-content/uploads/2015/06/PE_Ratti_Pianosa.pdf [accessed on 25 September 2022]). Sampling took place in spring 2017 on the Pontine Islands, and spring 2015 on Pianosa. Traps were placed randomly on the islands and set to be operational overnight, for a total of 5 trap nights per site. Captures were collected in the morning and stored at −20 °C until necropsy. Carcasses were thawed overnight at room temperature before necropsy, during which sex, age class, and external/internal abnormalities were recorded. Body condition was determined for each animal using the linear regression of body mass on total body length (tip of the nose to anus) [51]. To avoid cross-contamination, a sterile scalpel was used to collect each sample. 

Total genomic DNA was extracted from each sample of the spleen (≈10 mg) [52], skeletal muscle (≈25 mg of quadriceps femoris), kidney (≈25 mg) and central nervous system (CNS) (≈25 mg of brain homogenate) [43] using a commercial kit (GenEluteTM Mammalian Genomic MiniPrepKit, Sigma-Aldrich, St. Louis, MO, USA), according to the manufacturer’s instructions. 

Spleen samples were tested for *Anaplasma* spp./*Ehrlichia* spp. PCR analysis was performed with the primer pair PER1/PER2, amplifying a 452 bp portion of the 16S rRNA of the *Ehrlichia-Anaplasma* group [53]. Spleen samples were tested for *Babesia* spp. with a PCR protocol targeting the V4 hyper-variable region of the 18S ribosomal RNA gene [28]. A constant and specific fragment of *L. infantum* kDNA was amplified from spleen samples using an mRV1–mRV2 primer pair [52]. The skeletal muscle, kidney, and CNS were tested for *N. caninum* and *T. gondii*. *N. caninum* DNA was detected using the Nsp6Plus–Nsp21Plus primer pair [43]. A specific 575bp-long fragment of the *T. gondii* ITS1 region was targeted using primers TOX3- and TOX4 [43] with regard to the samples from Pianosa, while samples from the Pontine Islands were analysed through the LAMP protocol described in [54], which amplifies a fragment of the SAG2 gene. The two methodologies were proven comparable in establishing the parasite prevalence [54]. Positive and negative control samples were included in each PCR assay and standard precautions were taken to avoid contamination. PCR positive samples were purified and sequenced (BMR Genomics, Padua, Italy). Obtained sequences were compared to the ones available in the GenBank to confirm the parasite’s identification. Only sequences returning with 100% identity and cover with those found in GenBank deposits are reported in this study.

### 2.3. Statistical Analyses

Prevalence calculations and statistical analyses were performed using the R version 4.2.1 [55]. The statistical tests used were as follows: chi-square test to compare the frequency of infection on the Pontine Islands; logistic regression implemented using the *glm* function in the *stats* package with a binomial family and *logit* link, to investigate the significance of sex, weight, and other host biometric variables in determining individual infection risk.

## 3. Results

The total number of rats sampled was 53, including 38 on the three Pontine Islands (7 on Ponza, 18 on Ventotene and 13 on Palmarola), and 15 on Pianosa (Table 1). The male-to-female ratio was balanced (Table 1), and no relevant abnormalities were recorded. No rodents were found in poor body conditions. Five individuals (13.6%) from the Pontine Islands were infested by the flea *Nosopsyllus fasciatus*, while two individuals (13.3%) from Pianosa presented ticks of the genus *Ixodes* (no further identification as it was beyond the scope of the study). 

### 3.1. Parasite Prevalence in Rats from the Pontine Islands

No individuals were found positive for *Anaplasma* spp./*Ehrlichia* spp. or *N. caninum* (0.00%; CI 95% 0.00–9.18%), while 14 rats, mainly trapped on Ventotene and Palmarola, tested positive for *Babesia* spp./*Theileria* spp. (36.84%; CI 95% 23.38–52.72%) (Table 1). However, there was no significant difference between the sites ($\chi^{2}$ = 0.25, df = 2, *p* = 0.88). All the sequences obtained were *Babesia* spp., belonging to the species *B. microti* and *B. microti*-like (100% homology and coverage with GenBank sequences). Logistic regression revealed that sex, weight, and other biometric variables were not significant in determining the infection risk. *L. infantum* was found in the spleen of two rats (5.26%; CI 95% 1.46–17.29%), both females from Ventotene (Table 1). *T. gondii* was found in the samples from 16 individuals (42.11%; CI 95% 27.85–57.81%) (Table 1). The skeletal muscle rather than CNS and kidney was the most reliable sample to detect positivity. In this case, the prevalence was higher in individuals from Palmarola and Ponza, but the difference in site prevalence was not significant ($\chi^{2}$ = 0.16, df = 2, *p* = 0.92). Similarly to *Babesia*, logistic regression revealed that no potential risk factor taken into account in the study was significant.

### 3.2. Parasite Prevalence in Rats from Pianosa

A single rat was found to test positive for *Anaplasma* spp./*Ehrlichia* spp. (6.67%; CI 95% 0.00–19.73%) (Table 1); sequencing results showed 100% homology and coverage with *A. phagocytophilum*. This rat was also positive for *Babesia* spp. (sequence showed 100% homology and coverage with GenBank sequences). Out of the 15 rodent spleens tested, 12 tested positive for *Babesia* spp./*Theileria* spp. (80.00%; CI 95% 59.05–100.00%) (Table 1). PCR positive samples were purified and sequenced, and the sequences included *B. divergens* (1 sample), *Theileria* sp. (1 sample), and *B. microti*/*B. microti*-like (10 samples). *Leishmania* sp. positive samples numbered 4 (26.67%; CI 95% 3.50–49.83%), of which sequencing confirmed to be *L. infantum* (Table 1). No individuals were found to test positive for *N. caninum* or *T. gondii* (0.00%; CI 95% 0.00–19.73%) (Table 1).

## 4. Discussion

In the context of black rat eradications on small islands, funded through European LIFE programmes, individuals were sampled on four small islands in central Italy, off the coast of Tuscany and Latium, three of which are part of the Pontine Islands (Ponza, Ventotene and Palmarola) and one belongs to the Tuscan archipelago (Pianosa). The number of individuals captured was comparable among the four islands and consistent with other trapping sessions conducted in the context of those rat eradication projects. In general, sampled rats exhibited a balanced male-to-female ratio and showed no signs of malnutrition or abnormalities.

No individuals from the Pontine Islands tested positive for *Anaplasma* spp., while only a single rat from Pianosa tested positive. Subsequent screenings reported no positive samples for *Anaplasma* spp. [56]. In Europe, *Anaplasma* spp. displays a great variation in prevalence among rodents and associated ticks (e.g., [39,57,58,59]). Its persistence seems to be determined by the presence of ungulates, considered the main reservoir hosts [35,58,60,61], and availability of preferred rodent and tick species [59,62]. However, similarly to what has been hypothesised for *Borrelia* sp. transmission, bird species may play a role in introducing and sustaining infection [58,63]. Hence, it is plausible that the positive rat on Pianosa may have been exposed to an infected tick carried by a migrating bird [64]. An alternative explanation for the absence or the low prevalence of these microparasites may be short-lived rodent infections [60,65]. In England, ticks, rather than rodents, seem to maintain the infection over winter, with high seasonally detectable rodent infections in summer and autumn [65]. This is associated with seasonal peaks of *I. trianguliceps* (small mammal specialist tick) nymphs and adults [65]. In this study, black rats were sampled in spring, and rodents may have already cleared the infection. Nevertheless, this cannot be confirmed and needs further investigation through sampling rats and relative ticks in different seasons on islands where eradication is still underway. Due to the very low prevalence of *Anaplasma* spp. recorded in hosts, priority should be given to tick sampling and screening as a better strategy to assess the zoonotic risk.

High positivity in all locations was found for *Babesia* spp./*Theileria* spp., with the highest prevalence of 80% on Pianosa. Interestingly, later rat screening on Pianosa showed no *Babesia* presence among the sampled rats [56]. On the Pontine Islands, the positivity was higher (although not statistically significant) on the more isolated islands, Ventotene and Palmarola. The reason may have been the presence of a goat population, able to amplify the tick population and increase transmission, or the abundance of migrating birds, potentially introducing infected ticks. Comparable high prevalences of *B. microti* were reported in field voles, where the infection is usually sub-clinical and persistent, with individuals remaining PCR positive for years [66,67]. Our sequencing results identified strains of *B. divergens*, *Theileria* sp., and *B. microti* on Pianosa while on Pontine, only *B. microti* and *B. microti*-like were recorded. These species are known to have records of zoonotic infections [30,68,69].

In Europe, *B. divergens*, whose main reservoir is cattle, is traditionally considered the major *Babesia* species of zoonotic relevance. Recently, the zoonotic significance of the rodent specific *B. microti* complex has been reassessed due to its increasing involvement in human babesiosis [69]. Cattle have been absent from Pianosa for 20 years, so we hypothesise that the rat found infected by *B. divergens* may have acquired it through a tick transported by a migrating bird. Alternatively, rats or other species occurring on the island such as hares (*B. divergens* has been isolated in Lagomorpha, e.g., [70]) could represent competent reservoir for this *Babesia*, able to maintain an autonomous cycle. Further investigation into ticks and alternative host species (e.g., xenodiagnoses) is needed to better understand host–vector–parasite associations. To our knowledge, this is the first record of *B. divergens* recorded in a rodent, although it was reported to be cultured in rat erythrocytes [71]. This is a significant and novel finding of this study, although no further speculation can be made on epidemiological dynamics. The sequence identified as *Theileria* sp. was not identified at a species level, but domestic and wild ungulates, a common reservoir of the *Theileria* species [72], were not present on Pianosa. It is hypothesised that an infected tick was introduced by a migrating bird, but further investigations are needed to draw definitive conclusions. *B. microti* and *B. microti*-like were reported in rats from all locations, posing a potentially significant zoonotic risk. More in-depth investigations of *Babesia*-vector dynamics are needed to assess this risk, as the parasite is mainly vectored by the small mammal specialist *I. trianguliceps*, whose ability (and frequency) to feed on humans is still debated [73].

On the Pontine Islands, *L. infantum* was recorded in ~5% of sampled rats, while prevalence of infection on Pianosa was five times higher. The presence of the protozoan at such a low prevalence in black rats from Pontine, compared to Pianosa, was somewhat unexpected. Firstly, the main reservoir is present (although limited) on the Pontine Islands, and secondly, these islands are in the proximity of hyperendemic areas [74,75]. On Pianosa, where black rats exhibited a much higher prevalence, dogs are absent. Analogously, on Montecristo, another island of the Tuscan archipelago targeted for rat eradication, high prevalence was recorded despite the absence of dogs [25]. Further investigations are needed to confirm the existence of an autonomous cycle of *Leishmania* on Tuscan archipelago islands, where the introduction of infected rats in recent years can be ruled out. A larger sample size and vector sampling are essential to evaluate the actual parasite circulation as well as xenodiagnostic studies to ascertain black rats as competent reservoir hosts [76,77,78]. Nevertheless, it has been observed that other species, including rats, are able to sustain a *Leishmania* sylvatic cycle, in the absence of or in the presence of a limited numbers of dogs [25,40,41]. Rodents have also been found to be responsible for the emergence of new foci of cutaneous leishmaniosis in different countries [79,80,81].

*T. gondii* was recorded at a high prevalence on Palmarola and Ponza, but no rats from Pianosa tested positive. This may corroborate the absence of the protozoan from isolated islands of the Tuscan archipelago (see [25]). On the Pontine Islands, the high prevalence suggests a wide circulation of the protozoan, although the mechanisms of transmission could not be investigated and this study could not link directly the infection status of cats and rats. The high parasite circulation may be due to the dynamics between black rats, intermediate hosts, and definitive hosts, i.e., cats. On a small island in Japan, the life cycle and sexual reproduction of *T. gondii* seemed to be accelerated by predator–prey interactions between cats and black rats, resulting in high local infection levels [82]. Alternatively, high local infections may be due to the abundance of other intermediate hosts capable of amplifying the infection, such as migratory birds. These are recognised as intermediate hosts of *T. gondii* [83,84] and are heavily preyed upon by black rats [8]. High genetic variability of this parasite observed in black rats from an island in Brazil demonstrated its high capacity for adaptation in an insular environment, which may influence virulence and pathogenicity [26]. Interestingly, Pianosa (as well as Montecristo [25]), where no *T. gondii* was recorded, is an equally important roosting site for seabirds and other migratory species [14,17]. In the Tuscan archipelago, the possible role of wild birds in *T. gondii* epidemiology remains to be evaluated. Different bird species assemblages and their relative abundance in different islands may determine these differences in parasite transmission. Identification of the circulating genotypes may clarify the geographic distribution and hosts specificity to formulate hypotheses on introduction and transmission pathways [85]. *N. caninum* was not recorded in any rat sampled in this study, although dogs, the common reservoir of this parasite, are present on the Pontine Islands. The reason could not be ascertained, as the main reservoir and other wildlife species considered able to host this protozoan [86] were not tested. Thus, the dynamics of *N. caninum* in our study area need to be further investigated.

In this study, two cases of co-infection were reported. A single rat from a Pontine island was co-infected by *B. microti* and *Borrelia burgdorferi* sensu lato (s.l.), namely the aetiological agent of Lyme disease (LD) in humans (results of *B. burgdorferi* s.l. infections were not presented in this study as they are part of another epidemiological screening). The second co-infection case was found on Pianosa, where the rat infected by *Anaplasma* was also positive for *B. microti*. The first co-infection is widely reported in the literature, and it has been linked not only to an increase in transmission and emergence of *B. microti* in the enzootic cycle, but also to a greater disease severity and duration of LD in humans [87,88]. Co-infections involving *A. phagocytophilum* are considered generally less common, and in small mammals co-infections with *B. microti* were not significant [89]. Nonetheless, the association with *B. microti* has been previously reported in *Ixodes* ticks [90], in *Rattus* [91], and in humans [92]. Concurrent circulation of multiple parasites (or pathogens) in host populations is common in nature and within-host dynamics affect parasite virulence [93]. Parasites may have agonistic or antagonistic effects, determining different outcomes in terms of infection patterns, increasing or decreasing each other’s prevalence at the population level [94]. Comprehending co-infections (and antagonistic interactions) is imperative because understanding microparasite dynamics leads to improved prediction and control of parasites and disease within natural populations [95]. In the case of a small, isolated island invaded by an alien species capable of disrupting host– (vector) –parasite dynamics, understanding co-infections is key from a conservation, public health and veterinary point of view. Parasites introduced with invasive hosts may spread to native species, interacting with native parasite/pathogens (and vice versa), increasing disease risk; and this effect may have the potential to persist in the system after the eradication of invasive species [27].

## 5. Conclusions

All protozoa documented in this study present a potential zoonotic risk, underscoring the significance of such screening not only from ecological and veterinary perspectives but also from a public health standpoint. Our primary findings provided support for the hypothesis of short-lived infections of *Anaplasma* spp. in wild rodents. We reported the first instance of *B. divergens* in wild rats and provided evidence for the presence of an autochthonous *L. infantum* cycle in the Tuscan archipelago islands. Additionally, we observed the absence of *N. caninum* across all islands, even in areas where dogs were present.

While our results provided valuable insights, they would have benefited from a larger sample size, particularly through sampling campaigns conducted throughout various seasons to capture short-lived infections like those from *A. phagocytophilum*. Furthermore, for future assessments of zoonotic risk, it is advisable to sample vectors and other potential hosts. Despite the relatively small sample size, co-infections were detected. These should be thoroughly investigated within the context of invasive species eradication on island ecosystems because introduced generalist parasites that can be shared between invasive and native hosts may affect the latter both before and after eradication [3,27].

This study reaffirms the theory that invasive species, such as rats, could be accountable for sustaining an elevated parasitological threat to fauna and human communities in specific ecosystems [24]. This concern may be further exacerbated by the current backdrop of environmental changes, including climate change, which can alter the distribution of parasites and their vectors [96]. Consequently, this study includes useful knowledge on parasite geographical distribution and host interactions, offering key insights to inform public health and wildlife management policies.

## Figures and Tables

**Figure 1 animals-13-03279-f001:**
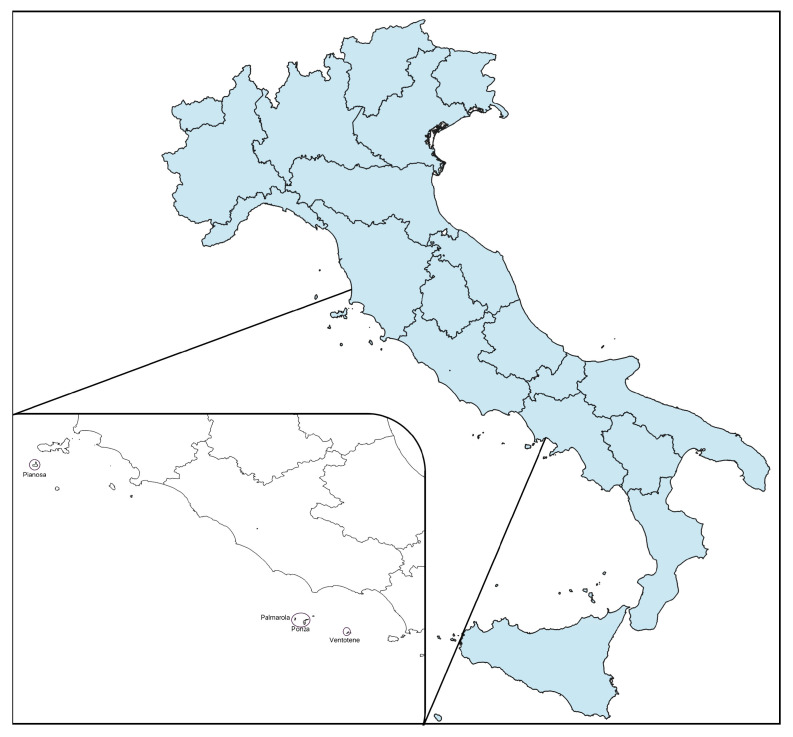
Map of study areas: Pianosa (Tuscan archipelago) and Palmarola, Ponza and Ventotene (Pontine Islands).

**Table 1 animals-13-03279-t001:** Summary of individuals sampled on each island, including absolute numbers of positive cases and the prevalence of the parasites of interest (95% confidence interval in brackets).

	*R. rattus*			Positive Samples—Prevalence % (CI 95%)				
	M	F	Total	*Anaplasma* spp.	*Babesia* spp./*Theileria* spp.	*Leishmania* sp.	*N. caninum*	*T. gondii*
Palmarola	7	6	13	0–0.00 (0.00–24.71)	5–38.46 (10.93–65.99)	0–0.00 (0.00–24.71)	0–0.00 (0.00–24.71)	6–46.15 (19.22–74.87)
Ponza	3	4	7	0–0.00 (0.00–40.96)	2–28.57 (0.00–64.72)	0–0.00 (0.00–40.96)	0–0.00 (0.00–40.96)	3–42.86 (9.90–81.59)
Ventotene	11	7	18	0–0.00 (0.00–18.53)	7–38.89 (15.71–62.06)	2–5.26 (1.46–17.29)	0–0.00 (0.00–18.53)	7–38.89 (17.30–64.25)
Pianosa	8	7	15	1–6.67 (0.00–19.73)	12–80.00 (59.05–100.00)	4–26.67 (3.50–49.83)	0–0.00 (0.00–19.73)	0–0.00 (0.00–19.73)

## Data Availability

Not applicable.

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
