# Peer review of "Zoonotic Microparasites in Invasive Black Rats (Rattus rattus) from Small Islands in Central Italy"

_animals, 2023, doi:10.3390/ani13203279_

Round 1
Reviewer 1 Report
The authors presented microparasite infection of an invasive species (black rat) in island circumstances. The investigation counts on the interest of the scientific community. The issue is current and has a high veterinary public health concern.
After minor, mostly grammatic, revision, the manuscript is strongly recommended to publish.
Major notes:
In the Discussion and Conclusion chapter, plenty of sentences are very wordy and difficult to follow.
All methods, which provided results should be descibed in the Materials and methods chapter.
The first half of the Conclusion should be moved to the Discussion chapter.
Minor notes:
Line 29, 42, and 107: 'rat role' should be replaced by 'role of rats' 'rat's role' rats' role' or other term, which sound more English
Line 95-97: This sentence means that the parasite has higher genetic variability than the hosts. I deem that the authors' intention was to compare parasites of different origin.
Line 129-130: 'host of vectors' sounds strange
Line 148: By common mention of T gondii and N. caninum causes misunderstanding, as if canines could be definitive hosts of T. gondii, which is not true.
Line 154-156: Hardly understandable sentence.
Line 168: 'covered in' should be replaced by 'covered by'
Line 167-171: Very long, hardly followable sentence.
Line 173: 'west from Ponza' is suggested to replace by 'west of Ponza'
Line 180-182: missing part of sentence
Line 245: 'while 15 in Pianosa' is incomplete clause
Line 256 and 263: In the Materials and method section, there was not any mention of logistic regression. Please, describe it there.
Line 261: comma after 'kidney' is unnecessary
Line 262: Please, replace 'e' with 'and'.
Line 263: in 'Logistic' unnecessary upper case
Line 281-282: This sentence needs completion. It is difficult to understand that it refers to others' study.
Line 290-291: 'highly seasonal detectable rodent infection' is suggested to replace by 'high seasonally detectable...'
Line 293-296: very long sentence
Line 300-302: The initiation 'Regarding Pontine island' is confusing. The sentence is difficult to understand.
Line 305, 308, 327: Maybe present perfect tense is not the best choice.
Line 314, 375-376: The term 'in alternative' needs a noun after it.
Line 329-332: Hard to understand.
Line 334-337, 339-342, 342-345, 364-366, 390-393, 405-408: Very long and/or complicated sentences, which are suggested to rewrite.
Line 350: Montecristo was not mentioned in the MatMeth section.
Line 361-362: This part of the sentence is hard to understand.
Line 401: 'N. caninum resulted...' sounds strange
Line 410: A comma is needed after 'ecosystem'.
Line 418: 'parasite vector distribution' should be replaced by a more English term
Most of the suggestions referred to the language. Please, see above!
Reviewer 2 Report
Dear Authors,
Your manuscript entitled “Zoonotic microparasites in invasive black rats (Rattus rattus) from small islands in central Italy”, provides crucial information about the presence and prevalence of important pathogens in rats in the confined environment of small Italian islands. The article is overall well-designed and well-written. However, you may find my comments for revising your manuscript useful. Please find my suggestions below.
Major
1. The part included between lines 119 and 156, in my opinion, would better fit in the Discussion than in the Introduction.
2. Line 199. The rodent species is Mus musculus or Mus musculus domesticus (a subspecies). Mus domesticus is not correct.
3. In parallel to the text, the results need to be presented in a Table with all the details of the location (Island) pathogen species numbers and percentages.
4. Line 266. What species of Anaplasma/Ehrlichia did the sequence show?
5. Line 269. “included B. divergens, Theileria sp., and B. microti.” How many samples per species?
6. Lines 324-326. The Authors declare here that "in this study" a rat was co-infected by B. microti and B. burgdorferi (s.l.). However, the latter pathogen was not examined "in this study" (accordingly, the Authors note that the "results are not presented here"). In my opinion, this result should not be discussed as a result of the present study.
7. Line 342. Could there also be a negative interaction in some cases? In this theoretical analysis, I suppose that this scenario should also be considered.
8. Lines 347-348. Why is it so unexpected? Here references to similar studies and their results (prevalences) are missing.
9. Line 357. Why mention specifically the cuteneous of all leishmanioses?
10. I believe that the Conclusions should be re-written with an effort to give them a more robust and targeted character. In their present form, the conclusions include a lot of repetitions of the Discussion. Actually, in some cases, whole sentences are almost identical to corresponding sentences of the Discussion (e.g. lines 398-401).
Minor
1. Lines 116-118. I would place this sentence before the previous, i.e. before stating which pathogens were addressed in the present study.
2. Line 132 I would suggest using the term “rickettsiosis” instead of “anaplasmosis”
3. Line 360. This information is not clear. How is this being ruled out? Also, why do the results support the presence of autochthonous leishmaniosis in Pianosa? Isn’t the same true for Pontine?
4. Line 363. Please replace “islands” with “animals”
5. Line 351 Please add “of infection” after “prevalence”
6. Line 364. Please replace “do occur” with “are present”
7. I would suggest using the term “leishmaniosis” instead of “leishmaniasis” according to https://pubmed.ncbi.nlm.nih.gov/3201706/ and in consistency with the rest of the infections’ names used in the manuscript (e.g. babesiosis).
8. The following is only a personal point of view that I would like to share with the Authors, not an actual request for correction. I am a little bit reluctant that parasitologists may use the term "microparasite" to include protozoan parasites, and bacteria (and viruses). I think that we cannot place bacteria and viruses in the same group as protozoan parasites, despite the relatively recent “trend” that some biologists follow. I would prefer to use the classical separate terms (i.e. bacteria and protozoan parasites) and dismiss the term “microparasites” overall.
English language would benefit from editing from a native speaker.
Here are some suggestions from my point of view (I am not a native speaker)
Line 74. “where it represents” please change to “where the black rat represents”
Line 77. Replace “object” with “the target” (to avoid repetition of "object")
Line 79/ Replace “being” with “as rodents are”
Line 129 Replace “infection” with “pathogen”
Line 225. Replace “Similarly” with “Additionally”
Lines 337-339. Please rephrase for better English. For example: "Comprehending co-infections is imperative because understanding the dynamics of microparasite interactions leads to improved prediction and control of parasites and diseases within natural populations."
Line 339. Replace “In this context, small” with “In the case of a small”
Reviewer 3 Report
The abstract provides a concise summary of the study, including the research question, methods, key findings, and implications. However, specifying the sample size and demographic characteristics of the rat populations would strengthen the abstract's completeness.
The introduction provides adequate background information on the importance of studying invasive species and their potential role as reservoirs for zoonotic parasites. It references the relevant literature, but more specific details on the selected microparasites and their importance in the context of the study could enhance the introduction's depth.
The methods section provides sufficient detail on the procedures followed for rat sampling, DNA extraction, and PCR analysis. However, it would be beneficial to include more information on the sample size and any limitations of the study.Add specific data on the sample size and demographic characteristics of the rat populations studied.
The references are relevant and provide context for the study. However, there could be a more detailed discussion comparing the current findings with existing literature to highlight the novelty and significance of the results.
The sentence "Microparasites were selected because of their zoonotic and veterinary relevance." could be rephrased for clarity, such as "We focused on microparasites with zoonotic and veterinary significance."
The sentence "Phylogenetic studies are needed to shed light on infection dynamics." could be more specific. What kind of phylogenetic studies are suggested? Clarifying this would enhance the sentence's effectiveness.
"individuals were kill-trapped" → It's more appropriate to say "individuals were trapped and euthanized."
"influence of rodent and tick species" → "influence of rodent and tick species on the persistence of the infection"
"Protozoa recorded in the study represents a potential zoonotic risk" → "Protozoa recorded in the study represent a potential zoonotic risk."
The overall quality of English is good, but some sentences are complex and could be simplified for better readability. For instance, in the sentence: "B. microti was reported in rats from all locations, representing a potentially great zoonotic risk," you might consider simplifying it to: "B. microti was found in rats from all locations, posing a significant zoonotic risk."
The phrase "The absence or very low prevalence of Anaplasma spp. recorded in this study has been commonly observed; however, to investigate the actual circulation of the protozoan and therefore establish the zoonotic risk, it seems preferable to test the tick vectors of the area of interest" is a bit convoluted. It could be simplified for better clarity.
See above
Reviewer 4 Report
The article, “Zoonotic microparasites in invasive black rats (Rattus rattus) from small islands in central Italy” describes the findings of Anaplasma spp./Ehrlichia spp., N. caninum, T. gondii, Babesia/Theileria and L. infantum infection rates of 52 rats sampled from the targeted region. Although a relatively small sample of rats was used for the study, the paper is worthy for publication as a small communication but only after the authors have conducted major revision of the draft.
Major points:
Although the authors claim that PCR and sequencing were performed, no Genbank deposits of the specific sequences is apparent and no mention of their query cover or percentage identity is reported. This is really disappointing as the reader would like to know of the true percentage similarity of these findings to reported parasites or compare these results in other phylogenetic studies in the region. If possible, please add this information.
The entire article needs to be reviewed and corrected by a native English speaking scientifically orientated person in order to correct grammatical errors that are evident throughout the text.
Minor points:
Line 26: Delete zoonotic in parentheses.
Line 30: “…different body tissues…Delete “(eg. spleen)”
Line 77-78: “targeted in several eradication programs, including on small islands of central Italy, the region of focus in this study [17,18]”
Line 92 and elsewhere throughout the manuscript, do not place the word “in” before the words “an island”. It must be written as “on Christmas island”
Lines 121-122: “Babesia spp. are among the most common parasites found in mammals’ blood [31], with reports of zoonotic infections.”
Lines 143-146:” Even if the black rat is a competent reservoir host for L. infantum, since recent studies support the hypothesis of sylvatic cycles of leishmaniasis independent from dogs, the latter are still considered to be the main reservoir of infection [25,44].
Line 150: “ or for other potential domestic or wildlife animal fauna”
Line 154: Remove “Whereas”
It would be better to place the lines 104-118 at the end of the section in order to introduce the reader to the objectives of the study in a more targeted way.
Line 192: What are “convicted residing”? Maybe, previous convicted prisoners who are now residents of the island?
In the discussion section, preferably begin with the significant positive findings of the study instead of negative or very low, i.e. Anaplasma spp. prevalence observed. Target it from what was new/novel and most surprising in the study towards those parasites not observed and lastly, the limitations of the study.
Similarly, in the conclusion, just give the take home message and implications from the study.
The entire article needs to be reviewed and corrected by a native English speaking scientifically orientated person in order to correct grammatical errors that are evident throughout the text.
Round 2
Reviewer 2 Report
Dear Authors,
Thank you for taking into consideration my comments. I have no further suggestions and I believe that your manuscript is suitable for publication.
Author Response
Thank you very much for your comments and suggestions. We believe it greatly improved the quality of the manuscript.
Reviewer 3 Report
Ensure sentences do not start with numbers or abbreviations.
Include important results in the abstract regardless of word limit.
In line 46, write 'Babesia divergens' in full before using the abbreviation 'B.' throughout the text. Apply the same rule for other species.
Provide details about sample size determination and sampling method.
Conclusions should summarize your results, study limitations, benefits, and future recommendations.
Moderate editing of English language required
Reviewer 4 Report
The authors have attempted to improve the manuscript, and certain parts, eg. Introductory paragraph that targets towards the aims of the work, the addition of a Table, the targeted conclusion have been improved substantially. However, the English grammar throughout the manuscript is still not suitable for publication purposes. It is not enough to replace each "in" an island to "on" an island for the entire sentence to be grammatically correct. I again strongly advise revision on the English grammatical style. Again, there are many mistakes, only some of which I have addressed below, that need to be corrected.
Line 54: Never start a sentence with a number. It should be written as " Fifty three"
Line 90: Is it not "Spirillum minus?"
Line 167: What are " domestic species assemblages?"
Lines 172-175: "Thus, microparasites of veterinary and zoonotic relevance selected for screening purposes included Babesia spp., Theileria spp., Anaplasma spp., and Ehrlichia spp., all the previously mentioned parasites transmitted by ticks; L. infantum transmitted by sand-flies; and the directly transmitted T. gondii and Neospora caninum coccidia.
Line 221: Remove " farming"
Lines 259-260: "Only sequences returning with 100% identity and cover with those found in GenBank deposits are reported in this study".
The table should also be aligned better within the text.
The English grammar throughout the manuscript is still not suitable for publication purposes. It is not enough to replace each "in" an island to "on" an island for the entire sentence to be grammatically correct. I again strongly advise revision on the English grammatical style. Again, there are many mistakes, only some of which I have addressed below, that need to be corrected.
